# Identifying adolescents' gaming preferences for a tobacco prevention social game: A qualitative study

**Georges Elias Khalil**[1]*, **Jeanie Kim**[2¤a], **David McLean**[1☉], **Erica Ramirez**[1☉¤b], **Bairu Zhao**[1], **Ramzi G. Salloum**[1]

**1** Department of Health Outcomes and Biomedical Informatics, University of Florida, Gainesville, Florida, United States of America, **2** Department of Behavioral Science, University of Texas MD Anderson Cancer Center, Houston, Texas, United States of America

☉ These authors contributed equally to this work.
¤a Current address: School of Medicine, Loma Linda University, Loma Linda, California, United States of America
¤b Current address: Malcom Randall Department of Veterans Affairs Medical Center, Gainesville, Florida, United States of America
* gkhalil@ufl.edu

**Data Availability Statement:** All data files is available from the Inter-university Consortium for Political and Social Research database, under the National Addiction and HIV Archive Program

## Abstract

### Introduction

Considering the dangers of adolescent tobacco use, the successful design of behavioral programs is crucial for tobacco prevention. According to preliminary research, social game interventions can improve adolescent tobacco outcomes. The current qualitative study aims to (1) uncover the gaming elements that adolescents deem important for a positive learning experience, and (2) confirm these gaming elements with adolescents who are presented with a tobacco prevention game concept that applies these elements.

### Methods

Findings from this study are drawn from two phases. Phase 1 involved in-person focus group discussions (n = 15) and Phase 2 included three online focus groups and a paired interview with another set of adolescents (n = 15). The study was conducted under a project that aimed to design and test a social game-based tobacco prevention program for adolescents (Storm-Heroes). With open coding and thematic analysis, two research team members identified repeated topics and relevant quotes to organize them into themes. The themes evolved as new content was identified during the process. This process was repeated until thematic saturation was reached.

### Results

Thematic analysis across Phase 1 and Phase 2 revealed four major themes: 1) Balance during gaming challenges, 2) Healthy social interaction, 3) Performance and creative freedom, and 4) Fictional world and game mechanics for tobacco prevention.

(NAHDAP-191749). Data requests may be sent to the University of Florida research team at gkhalil@ufl.edu.

**Funding:** Research reported in this publication was supported by the National Institute on Drug Abuse of the National Institutes of Health under award number R00DA044277, award received by GEK. The funder website is: https://nida.nih.gov/ The content is solely the responsibility of the authors and does not necessarily represent the official views of the National Institutes of Health. The funders had no role in study design, data collection and analysis, decision to publish, or preparation of the manuscript.

**Competing interests:** The authors have declared that no competing interests exist.

## Conclusion

This study identified specific intervention features that best fit the needs of adolescents in the context of a social game for tobacco prevention. For future research, we will use a participatory approach to allow adolescents to take part in the design process, improve Storm-Heroes, and develop health promotional messages that can be incorporated into the program. Ultimately, a board game for tobacco prevention is expected to bring adolescents together to create lasting memories that nudge them away from tobacco use and the harm it can cause.

## Introduction

### Background

More than five million high school students and more than 1 million middle school students in the United States reported using a tobacco product at least once, according to a 2021 national survey [1]. Among adolescents, tobacco use in its many forms has been linked to nicotine dependence [2], psychiatric disorders [3], and early signs of pulmonary disease [4]. The successful design of tobacco prevention programs is crucial in the fight against tobacco use at a young age.

According to previous reviews of tobacco preventive interventions, the application of social influence and social competence components has been shown to be most effective in promoting both the prevention and treatment of adolescent tobacco use [5, 6]. Behavioral interventions that incorporate social influence and social competency into the curriculum have demonstrated a considerable impact at both short- and long-term follow-ups [6].

Playing social games can be a successful tactic for fostering positive social influence and social skills. There is evidence that playing board games with others can encourage discussions about health [7–11]. Teens who participate in structured activities are more likely to develop peer group support, as they regularly interact and share goals and experiences through activities [12, 13]. This occurs when activities allow adolescents to interact regularly and share experiences and goals [12, 13]. In particular, multiplayer games can help to create a sense of peer group identity [14–16] as well as the shared values and norms that are associated with the activity [17–19].

According to preliminary research, social game interventions can improve both adolescent and adult tobacco outcomes. In one study by Khazaal and colleagues, adult participants who received a social board game exhibited lower rates of smoking frequency at a 3-month follow-up [20]. There is, however, a paucity of studies on the effectiveness of board games in preventing teen tobacco use. A recent study revealed that the board game "Smoke Stacks" can alter adolescents' attitudes toward tobacco use [21]. A single-arm pilot study of the board game "GiochiAMO" found that youths improved their knowledge about the dangers of tobacco use following gameplay [22]. Unlike conventional instructional techniques, board games are inventive, inexpensive, and more socially interactive. The uptake of board games is facilitated by their simple self-administration and low-cost implementation.

### Theoretical framework

According to the social learning theory [23], learning occurs through observed, imitated, and reinforced behavior. Social games can promote tobacco risk education by giving players the chance to witness and emulate healthy behaviors such as refusing tobacco offers. Additionally,

social games offer rewards for making healthy decisions, such as earning points for choosing healthy alternatives to tobacco use. In addition, by engaging in tobacco prevention content with others, adolescents begin to perceive norms against tobacco use and become motivated to model their peers with healthy behaviors.

### A human-centered participatory design approach

Human-centered design (HCD) is a method for designing services and products that are centered on the needs and challenges of a population. HCD advances participatory action research by creating solutions to problems with adolescents, rather than just taking notes of the problems that they report. By putting adolescents at the center of our research, the researcher becomes fully immersed in their issues [24–26]. Through this approach, adolescents are invited to be involved through participatory design. It is a cooperative process that strives to develop a health education program that is usable and beneficial for adolescent tobacco risk communication [24]. Similar to the work on the design of tobacco graphic warning messages [27], the participatory design technique can include adolescents in the design process to create a social game for tobacco prevention.

After-school youth organizations can provide a supportive environment for adolescents' participation in the design process and their engagement in a social game-based program for tobacco prevention. Previous research suggests that well-crafted activities within after-school organizations can promote youth empowerment [28], reduce substance use [29–32], and improve quality of life [33]. As a result, the design of an intervention that can be implemented within structured youth organizations may be an effective strategy for the prevention of adolescent tobacco use.

### Gap in knowledge

Currently, little is known concerning the best ways to improve the features of tobacco prevention interventions in order to boost their success. Recently, researchers have been designing and testing advanced game-based programs for tobacco prevention [34, 35]. Nevertheless, such programs have focused on direct human-computer interaction, and they may benefit from social features that can promote peer-to-peer interaction. On the other hand, researchers have also considered the development of socially interactive programs for tobacco prevention. However, such programs are currently missing game-based features that can improve adolescents' engagement.

### Study aims

In response to this knowledge gap, the objective of this qualitative study is to inform the design of a board game for tobacco prevention within after-school youth organizations. Particularly, we aim to determine specific intervention features that best fit the needs of this population. By leveraging qualitative research strategies, the study aims to (1) uncover the gaming elements that adolescents deem important for a positive learning experience, and (2) confirm these gaming elements with adolescents who are presented with a game concept that applies these elements.

## Materials and methods

### Study setting

Findings from this qualitative study are drawn from two phases. Phase 1 involved in-person focus group discussions (n = 15) and Phase 2 included a series of online focus groups and an

in-depth interview with another set of adolescents (n = 15). The study was conducted under a project that aimed to design and test a social game-based tobacco prevention program for adolescents.

## Inclusion/exclusion criteria

Adolescents between the ages of 11 and 18 years who were English-speaking and members of after-school programs were eligible for the study. With restrictions due to COVID-19 that prevented the convening of in-person meetings, adolescents were required to have access to a webcam, a computer, and the internet to participate online.

## Recruitment and sampling strategy

For Phase 1, participants were recruited from a pool of adolescents belonging to an after-school organization in Texas (the Boys and Girls Club). For Phase 2, participants were recruited from a pool of adolescents belonging to two youth organizations in Northern and Central Florida (the 4-H Program and the Boys and Girls Clubs) and a registry of potential research participants from underserved Florida counties through a community engagement program (HealthStreet). HealthStreet applies evidence-based community outreach practices to bridge the gap between community members and the health resources available to them, including participation in research [36, 37].

In this study, we approached 46 parents in person at youth organization sites, over the phone, or via videoconferencing. Parents of 39 adolescents approved of their children's participation and provided parental permission, and 30 adolescents participated in the study (15 participants per phase). During Phase 1, we had three focus groups with five participants per group. During Phase 2, two participants were randomly selected to take part in an in-depth paired interview, and thirteen participated in three focus group discussions that were composed of 3 to 5 participants per group. Two participants dropped out of Phase 2, leading to 28 participants completing the survey for the entire study (93.3% retained). We planned to recruit until information saturation was reached with 28 participants (Fig 1).

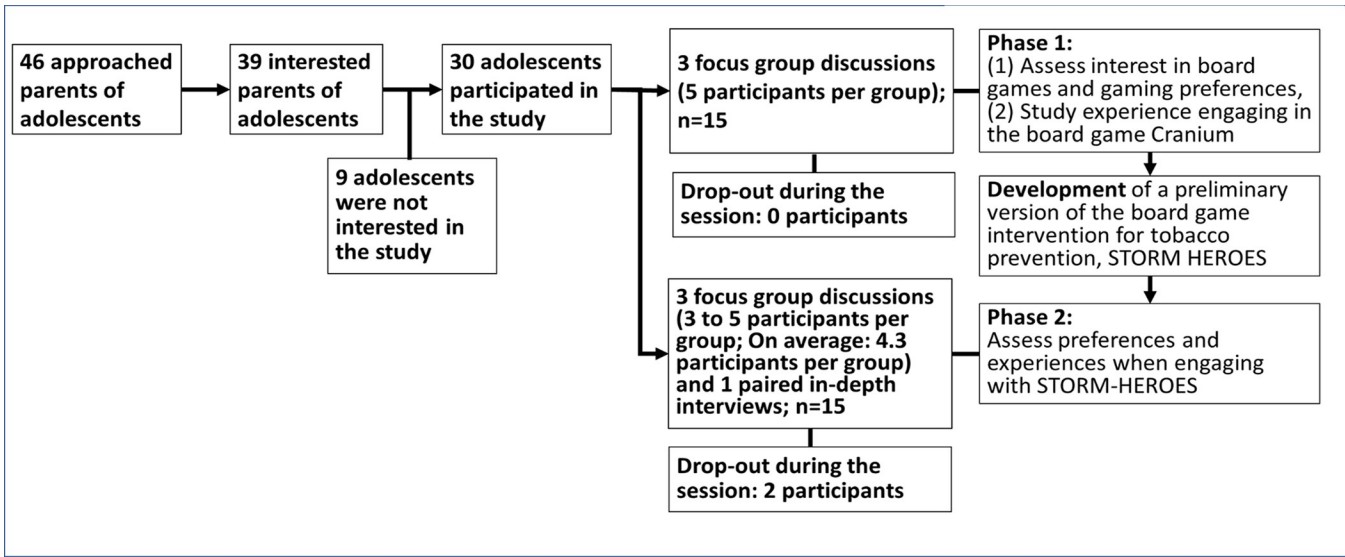

**Fig 1. Study flow diagram leading to the qualitative sessions.**

In Phase 1, qualitative sessions assessed adolescents' interest in board games, gaming preferences, and experiences engaging in the board game *"Cranium"* (described under the instruments section) [38–40]. Once adolescents' preferences were identified, we developed an early version of the board game intervention. Following game design, we conducted Phase 2 to evaluate adolescents' preferences and experiences engaging with the preliminary version of the intervention. While each phase was conducted separately, qualitative validation of the themes required merging the results from the two phases.

## Ethics statement

The current manuscript adheres to the consolidated criteria for reporting qualitative research (COREQ; S1 Table). Phase 1 of the study was approved by the University of Texas MD Anderson Cancer Center Institutional Review Boards (IRB) and Phase 2 was approved by the University of Florida IRB. Adolescents and their parents were informed of the study's purpose and procedure. They provided written informed parental consent and written informed child assent. Discussions for Phase 1 were conducted in person, while discussions for Phase 2 were conducted via Zoom video conferencing software. During remote discussions, participants were instructed to be in an environment without other people to keep the session private and ensure a safe and relaxed atmosphere. For both phases, participants were required to maintain the confidentiality of their own identity and the identity of other participants. Participants were given the chance to take breaks during the interviews. Participants were told they could withdraw from the study at any time. After a session, participants received a $10 gift card as compensation.

## Study instruments

For Phase 1, during each focus group session, the moderator asked participants a series of open-ended questions using a semi-structured instrument. We pilot-tested the semi-structured instrument with two young-adult volunteers and five adolescents and then revised it prior to the sessions. The instrument included a series of open-ended questions that ask adolescents about their preferences, attitudes, and beliefs regarding group activities, games, and board games (S1 File). Following the open-ended questions in Phase 1, participants played the game "Cranium" as a group, and then they provided their creative thoughts regarding the activities in the game. We chose "Cranium" because it is a multiplayer cooperative game that includes a variety of cognitive and creative social game-based activities across different domains [33–35]. Players work in teams in this game to complete activities, including charades, drawing games, sculpting with molding clay, and answering trivia questions (Table 1). Following gameplay, the moderator asked participants to share their opinion about the "Cranium" activities. Finally, the moderator presented participants in Phase 1 with an exercise that allowed them to use their imagination as they pitched ideas for a tobacco prevention board game. In stages, the moderator verbally described a scenario for a potential board game program, and participants were encouraged to elaborate on the features of this game (S1 File).

Between Phase 1 and Phase 2 of this study, we developed a proof of concept for the tobacco prevention board game, called Storm-Heroes, to be used for Phase 2. In this cooperative board game, players work together to protect an island from a storm that releases tobacco products and dangerous chemicals. Players choose to play as one of seven characters that live on the island. In this game, players move their pawns across the board, as they answer trivia questions, play mini-games, and work together to make the island healthy again. S2 File describes the gaming concept in more detail.

**Table 1. "Cranium" game activities and descriptions.**

| Activity | Description |
| --- | --- |
| Charades Game | One player acts out a word or phrase using only silent cues while the rest of the team tries to guess |
| Pictionary Game | One player draws a word or phrase while the rest of the team tries to guess |
| Sculpting Game | One player creates a sculpture using a modeling compound to make art while the rest of the team tries to guess |
| Miming Game | One player pretends to be a famous person using words and actions while the rest of the team tries to guess |
| Trivia Questions | The team answers multiple-choice and "true or false" questions |
| The Out-of-Place Game | The team picks two items that do not belong from a list of five words or phrases |
| The Spelling Game | The team spells a word backward, one per person at a time |
| The Definition Game | The team answers a multiple-choice question about the definition of a word |

For Phase 2, during each focus group session and paired interview, the moderator asked participants a series of open-ended questions using a semi-structured instrument. We pilot-tested the instrument with two young-adult volunteers and two adolescents, and then we revised it prior to the sessions. The instrument consisted of 48 questions that asked participants about their preferences concerning the narrative, characters, rules, and art of Storm-Heroes. Within each session, the moderator described the game and weaved in the questions from the instrument. Following the open-ended questions, the moderator asked the participants to play a series of activities from Storm-Heroes as a group. After playing the game, participants provided their feedback about their gaming experience. The instrument for the complete session can be found in S1 File.

For both phases, participants completed a survey that included questions about demographics, ever use of tobacco products (i.e., cigarettes, cigars, hookah, and vaping), and susceptibility to use tobacco in the future [41, 42].

## Data collection

For Phase 1, data collection involved one lead researcher (GEK, gender: male, credentials: M.P.H., Ph.D.) and one research assistant (JK, gender: female, credentials: B.S.). For Phase 2, data collection involved the lead researcher (GEK, gender: male, credentials: M.P.H., Ph.D.), a graduate research assistant (DM, gender: male, credentials: M.A.), and a research coordinator (ER, gender: female, credentials: B.S.). All interviewers were trained in qualitative research methods. There were no interviewer-related biases identified in this study.

Phase 1 took place face-to-face in a private room at the youth organization. Phase 2 sessions were conducted using the online video conferencing software Zoom due to COVID-19 restrictions, and participants joined the session from their electronic devices. To collect quantitative data, a paper-based survey was used for Phase 1 participants, while an online survey was distributed via REDCap for Phase 2 participants. During Phase 2, participants were instructed to be in a room by themselves for the entirety of the session. For both phases, an icebreaker was used at the beginning of the session to help establish a relationship between participants, and sessions took about 90 minutes.

## Qualitative data analysis

This study sought to identify intervention features that can support a successful learning experience, as reported by adolescent participants within and between Phases 1 and 2. Following

established qualitative research methods of thematic categorization, we chose to merge data from both phases as a means to identify overlaps and agreements between participants from both phases, thus reinforcing the validity of our findings [43]. This also allowed us to provide a comprehensive understanding of the themes that emerged across the entire study to capture both breadth and depth in the findings [44, 45]. A third-party transcription service transcribed the interview recordings. Two research team members analyzed the transcripts to identify emerging themes. We used open coding and thematic analysis to identify new and emerging themes. We analyzed each transcript to find repeated topics and relevant quotes, which were then organized into a list of themes. In line with grounded theory, the themes evolved as new content was identified during the process. We repeated this process until thematic saturation was reached. Considering that we applied an iterative human-centered design approach, the intercoder agreement was reached through a qualitative process. The two coders met on a regular basis to discuss the coding process in real time until they reached a full agreement. This method allowed the coders to discuss any disagreements and resolve them, instead of having a single intercoder reliability score with a limited opportunity for interpretation. We used Microsoft Word and Excel to manage and analyze the themes. These themes embody the results of the current study.

## Results

Phase 1 participants (n = 4; 26.67% female; mean age = 12.93 years, SD = 0.80) reported engaging in about twelve hours per week of gameplay, on average (SD = 14.15). All participants in Phase 1 self-identified as African American or Black, and three participants reported being susceptible to using tobacco in the future. In Phase 2, participants (n = 8; 57.14% female) had an average age of 14.71 years (SD = 2.33). Participants during Phase 2 reported engaging in about six hours per week of gameplay, on average (SD = 7.67). Seven participants (50%) in Phase 2 identified as African American or Black, and 5 participants (38.46%) were susceptible to using tobacco in the future. None of the participants ever used a tobacco product in any of the phases.

   The qualitative data scripts will be made available through a data repository. Thematic analysis across Phase 1 and Phase 2 revealed four major themes: 1) Balance during gaming challenges, 2) Healthy social interaction, 3) Performance and creative freedom, and 4) Fictional world and game mechanics for tobacco prevention. We examined these themes in the context of participants' experience with 1) games in general, 2) activities in the board game *"Cranium"*, and 3) activities within our developed board game Storm-Heroes. Fig 2 presents a conceptual framework that lists the identified major themes and their subthemes, driving a positive learning experience in a board game for tobacco prevention.

### Theme 1: Balance during gaming challenges

   **A balance between difficulty and capability to overcome challenges.**   After playing "Cranium", participants in Phase 1 expressed a preference for "challenging" game features and activities. Particularly, the presence of a timer, activities that require active thinking, and activities that involve hand-eye coordination were found to be particularly enjoyable because of their difficulty. Nevertheless, participants indicated that the difficulty of these activities must be in balance with their capabilities. In other words, they wanted to feel challenged only to the point that they could still feel in control.

- "I like how tense it ["Cranium"] is cuz like there's a timer. . . and that's challenging" (Phase 1, Focus Group 3, Participant 1)

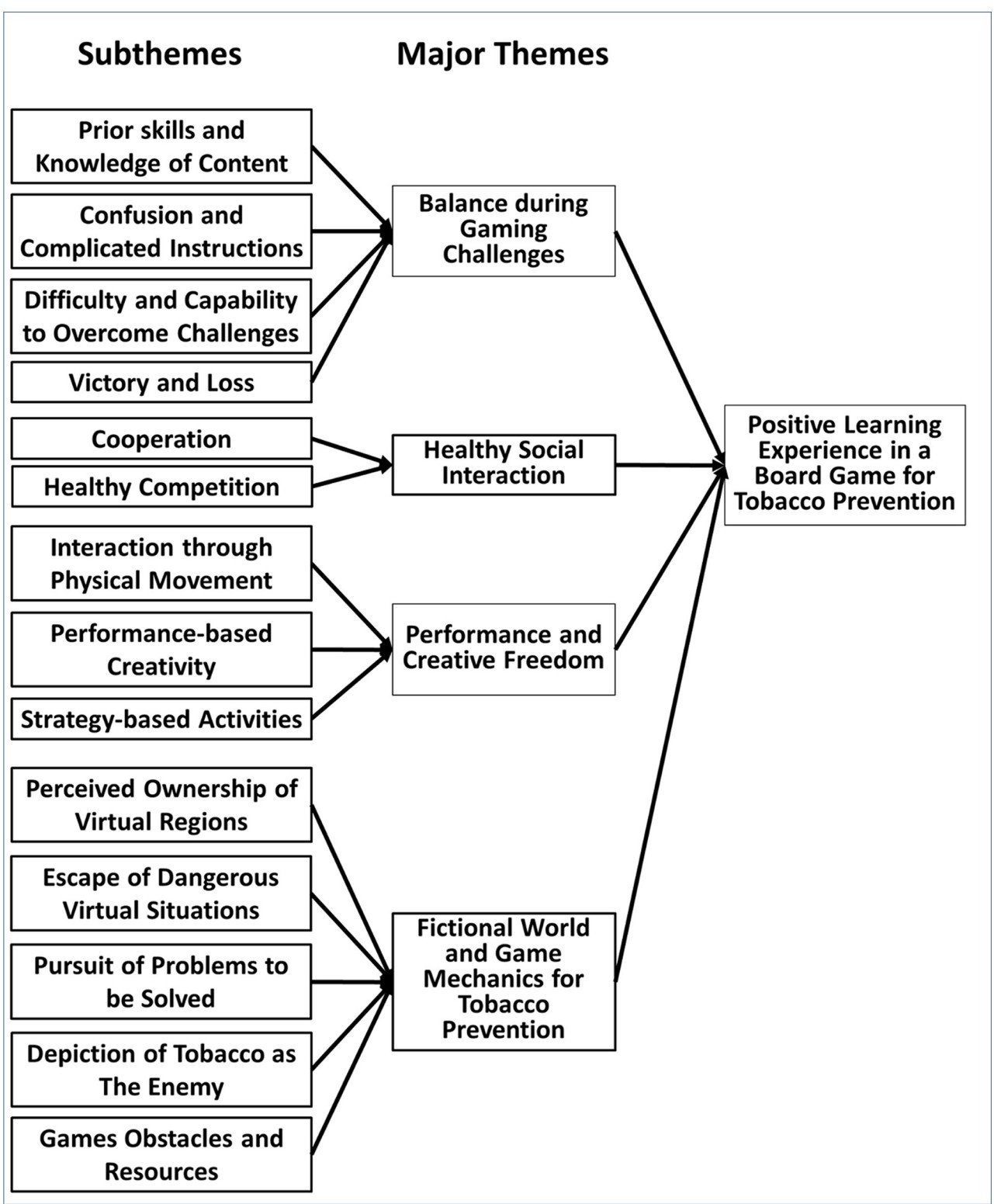

**Fig 2. Conceptual framework from identified study themes.**

- "Arranging the words because it's difficult. I like that it challenges your mind." (Phase 1, Focus Group 2, Participant 4)

- "Prefer drawing with eyes closed because it's more challenging." (Phase 1, Focus Group 2, Group response)

- "[My favorite "Cranium" activity] is not that hard but it's not like super easy" (Phase 1, Focus Group 3, Participant 2)

- "Yeah, my favorite was Odd Couple because—it's not that hard, but it's not super easy. It's kind of a challenge" (Phase 1, Focus Group 3, Participant 2)

Finally, even after playing Storm-Heroes, participants in Phase 2 also expressed enjoyment of challenging activities (e.g., the need to mime without talking to get others to guess a word). Participants preferred a game that presented enough of a challenge but only to the point that they can still achieve game success. For example, they expressed that it was "cool" that the game was "a little bit hard" but that one can "get it" right (Phase 2, Focus Group 1, Participant 4). They also explained that the activity is "not difficult like to where it will hinder the game" (Phase 2, Focus Group 4, *Participant* 2).

**The role of prior skills and knowledge.** Participants reported disliking activities that involved testing their knowledge about certain content. Particularly, trivia questions were often cited to be the participants' least favorite activities because they involved content that they had not yet learned. As a result, adolescents' attitudes toward certain activities were dependent on their prior set of skills or knowledge.

- "One thing I don't like about this game is like—there's some stuff on here we don't know about—like at all. Like one of those songs you had to—the song, Macho Man, that we had to hum, nobody here knew that song. I think they should make a new version—like 2019." (Phase 1, Focus Group 1, Participant 2)

- "My least favorite one was Trivia Questions because like—the question asked, "Which city's skyline was this?" and sometimes, you don't know which one it is, and you would have to really know skylines to know that question." (Phase 1, Focus Group 3, Participant 1)

- "Trivia was also my—my least favorite one because it's advanced stuff" (Phase 1, Focus Group 3, Participant 1)

- "[The Definition game] it gets complicated. . . Why don't you just say, "Look for this word in the dictionary." And then, if you—they give synonyms—I can't say it." (Phase 1, Focus Group 1, Participant 1)

**Confusion and complicated instructions hindering gameplay.** Although the participants welcomed being challenged during gameplay, two groups reported frustration when facing difficult activities in games during Phase 1 of the study. Several groups noted that "*confusing*" and "*complicated*" instructions and rules are the components that they generally disliked about board games. They expressed the importance of making activities "less complicated" (Phase 1, Focus Group 2, Participant 1), especially during gameplay. Participants reported the same experience with "Cranium", explaining that "reading the instructions", and encountering "confusion about the rules" were their least favorite experiences during the board game. Particularly, participants preferred to have the instructions coupled with "demonstration" and practice. *Although limited to the second focus group, confusion was also evident among participants who played Storm-Heroes during Phase 2 of the study.*

- "I don't like the fact that the other team, sometimes, whenever they have to read it (the card), it doesn't make any sense. . . the other team gets confused. . . they don't really know what they're talking about" (Phase 1, Focus Group 3, Participant 3)

- Group agrees that visual instruction/demonstration is easier to understand than written. (Phase 1, Focus Group 2, Group response)

- "I just need—I don't like the rules, like the paper. You have to read. I don't like that, I just like demonstrations. Like when you demonstrated to me then I just get it right away. "(Phase 1, Focus Group 2, Participant 4)

- "When I first played, it was pretty confusing then as I went along I kind of started to understand it." (Phase 1, Focus Group 2, Participant 1)

- I think it's great. I mean, I've had a lot of trouble like not understanding the rules. But these seem like simple and easy to understand and I like that. (Phase 2, Focus Group 2, Participant 5)

- And they're pretty set out to where there's no confusion. So then there's no arguments when something comes up. Like it's like this is how it is. There's no like changing that. (Phase 2, Focus Group 2, Participant 2)

   **The role of victory and loss after a challenge.**   Participants in Phase 1 expressed that achievement is important for their game enjoyment, preferring certain activities because they were "good at it" and expressing strong negative responses to loss during a game.

- "I like Spoons because I'm good at it" (Phase 1, Focus Group 3, Participant 2)

- "It gets frustrating when you don't get it right" (Phase 1, Focus Group 1, Participant 1)

- "When I lose [in board games]. . . it's terrible. It's like I just fell off a cliff or something. It's bad. I don't like it" (Phase 1, Focus Group 3, Participant 1)

## Theme 2: Healthy social interaction

   **General support for cooperation and competition.**   During Phase 1, when it comes to general game-based activities, participants expressed favorable experiences and opinions regarding social interaction. Teamwork and cooperation were conveyed favorably in all three groups of this study. According to participants' reports, they enjoyed working together and receiving help from one another during gameplay. They also identified social play as a crucial strategy to manage their emotions during gameplay. The same sentiment was expressed after playing "Cranium".

- "I like it [working with a team] because your teammates can help you get through difficult obstacles" (Phase 1, Focus Group 2, Participant 2)

- "Y'all can actually cooperate with each other and when you get mad they [teammates] can calm you down" (Phase 1, Focus Group 1, Participant 1)

- "[What I like most about it is] That you can play with your friends" (Phase 1, Focus Group 2, Participant 2)

- "What I like most about this game ("Cranium"), you got to. . . [play with]—the people on your team" (Phase 1, Focus Group 1, Participant 2)

   Although cooperation was a key component of their gameplay, participants shared their experiences with the competition. They tended to support a moderate level of competition,

teamwork, and cooperation. Yet, adolescents expressed dislike for excessive competition, explaining that it can evoke unfavorable responses (e.g., anger).

- "[What I like about social games is that] you can go against each other and you could try to win. Even if you lose, it's still fun" (Phase 1, Focus Group 3, Participant 1)

- "Because when you do these certain activities [social games], it gets you in a competitive and cooperative mindset. . . when you're being competitive, don't like make the competitive mindset take over you. . . that competitive mindset can turn into an anger outburst and that outburst can control you" (Phase 1, Focus Group 1, Participant 3)

**Cooperation in Storm-Heroes.**   First, participants in Phase 2 generally described Storm-Heroes to be "very interactive" and "fun to play" *(Phase 2, Focus Group 3, Participant 5)*. Participants generally liked the interactive activities, and they enjoyed taking turns between getting others to guess and being part of the group that is guessing.

- I think it's good, because it's very interactive. Like all three mini-games require the entire group to do something. Because one person is doing an action and the other people have to stay involved and like [be] attentive to what they're doing, so that they can get the right answer. So I think the involvement is really good. (Phase 2, Focus Group 2, Participant 1)

While playing the Storm-Heroes game, participants in Phase 2 agreed with Participants in Phase 1 by describing the advantages of working together to achieve success in the game. According to the participants, working together to overcome challenges is a key feature of cooperative gameplay. Participants agreed that this environment of cooperation fits well with the narrative of the game, in which they need to "save an island from a storm".

- "[What I liked most about this game is] how we have to work together and it's not as one person." (Phase 2, Focus Group 1, Participant 3)

- Oh, my bad. I said you can't do it by yourself. If you're on the island, then you might need help to do it (save it from the storm). If you got other people to help you, you might as well work as a group. You have more people to help and more minds. (Phase 2, Focus Group 4, Participant 1)

- Um, I mean, uh, they kind of seem to help out a lot, especially if you're working as a team. Because let's say like you have—you don't have the answer, but you have something that could help make the answer whole, then you could all put your answers together and come up with one like, big solution to the storm card. (Phase 2, Focus Group 4, Participant 2)

**Healthy competition in Storm-Heroes.**   When asked further about their opinions on working together, participants in Phase 2 expressed that the game Storm-Heroes should also include a competitive component. They described competition as "fun" and "challenging," saying that it positively adds to the collaborative features of the game. They explained that competition can still be "healthy" when everyone is winning, but to different degrees.

- I think that (competition) brings in, like, a lot of fun and like even though you're working as a team, because you all want to save the island, there's still healthy competition of, like, who can save it the fastest or the most. (Phase 2, Focus Group 2, Participant 1)

- I personally like a board game like Monopoly, because there's still that competition, but then there's still a sense, a little bit, of like you have to work together. (Phase 2, Focus Group 2, Participant 2)

- Because I think it gives the game a little more of a challenge to want to—Like you have to play against somebody who's also trying to like, make it worse. (Phase 2, Focus Group 4, Participant 2)

Participants also suggested ideas for incorporating competition into the game. Some participants suggested competition through the game activities. Particularly, as players draw cards and play activities, they can compete by counting the number of challenges they successfully overcome. On the other hand, a competition was suggested through the storyline of the game, by getting rid of "the most amount of chemicals" or having "the healthiest crop".

- Yeah, I liked it. I liked it. I think–I think once like as you're playing the game and different people are getting the Drawing Cards and stuff like that, I think that would be really fun. You know, I think about if everyone had a chance of drawing, you're going to have some people who are better than others, and I think that's part of what makes it fun. (Phase 2, Focus Group 2, Participant 1)

- I think that's (working as a team to save the island) good, but then if there could be, like, some type of competition. Like if you get rid of the most amount of the chemicals and like you have the healthiest crop, and that type of stuff. Like you kind of bring in like a Monopoly almost thing where you–there still is a winner and you're working collectively on a main goal. But there's still that first-place winner, so then you still have the fun part on the competition of a board game. ((Phase 2, Focus Group 2, Participant 2)

- I think L's trading idea earlier seems like it would be good. Like maybe you would have to go in a certain order to answer the questions. That way it would add a little bit of competitiveness, and you could, like, trade with the person who's being asked at that time. (Phase 2, Focus Group 2, Participant 5)

### Theme 3: Performance and creative freedom

**Interaction through physical movement.** After playing the social game *"Cranium"*, participants conveyed their interest in playing a variety of physically interactive activities (e.g., charades and drawing). Participants also described such activities as ones played with family members and friends. While participants express familiarity with static board games during which movement is solely on the board, they were surprised to play a board game that involved physical movement.

- "I liked it when he was describing the character and stuff, and he was drawing the artist— whatever he was drawing. It was like a family and friend game." (Phase 1, Focus Group 1, Participant 2)

- "What I liked about it is that it's more than I probably thought it was. Like I thought the game was just going to be sitting down and moving around your character but it's more of like a—more interactive game, like—"(Phase 1, Focus Group 2, Participant 3)

**Performance-based creativity.** As participants were playing the different activities of "Cranium", they expressed interest in the activities that involve performance and creativity. They found the Charades to be enjoyable, citing reasons such as loving "action" and finding "acting out things fun." In addition, some participants expressed interest in activities that allowed them to be creative and artistic. During Phase 2, participants who played Storm-Heroes shared similar preferences, including acting and miming games.

- "...when I understood what it was, I was like—it's like a Charades game using action because I love—I love action." (Phase 1, Focus Group 1, Participant 3)

- "Acting out things, like Charades and stuff, that was one of the main subjects that everybody kept talking about. We find that interesting." (Phase 1, Focus Group 3, Participant 1)

- "my most favorite was [miming]. ... it was pretty easy, and it was similar to Charades, so that's why I liked it the most." (Phase 1, Focus Group 3, Participant 1)

- "My favorite one was Sculpting because I like—like 3D art" (Phase 1, Focus Group 3, Participant 1)

- "I like the drawing part of the game "(Phase 1, Focus Group 1, Participant 2)

- "I think that the drawing, the acting and the speaking out would be cool, and the trivia" (Phase 2, Focus Group 1, Participant 3)

- It was, like, kind of–not like Pictionary but like–like you know like those games you have to act out the stuff? Then people would guess. So that's like–that's a fun game. So that's why I like it. (Phase 2, Focus Group 1, Participant 5)

**Enjoyment of strategy-based activities.** Although brief during Phase 1, some participants explained that they preferred to engage in challenging activities that involve strategy. To one participant, it is crucial to mentally put together a strategy to win a game.

- "I like Connect 4 because it's like strategies" (Phase 1, Focus Group 2, Participant 1)

- "[What I like about these board games is that] you have to have a strategy during the game..." (Phase 1, Focus Group 3, Participant 2)

## Theme 4: Fictional world and game mechanics for tobacco prevention

**Perceived ownership of virtual regions.** Some participants expressed a territorial mindset. They found it engaging to own the virtual world and its unique areas (e.g., beach, mountains, and caves).

- "What's cool about having an island, that's going to be your own place" (Phase 1, Focus Group 1, Participant 2)

- "Like not just only an island but different types of areas." (Phase 1, Focus Group 3, Participant 2)

**Escape from dangerous virtual situations.** Participants recommended that the challenge in the game occurs by escaping a dangerous situation (e.g., running away from an enemy).

- "ways for you to get caught by whatever monster's chasing you. And each turn, it moves one space or something, and there's cards, you have to answer questions relating to the topic—to go to certain places, just like checkpoints or something." (Phase 1, Focus Group 3, Participant 1)

- "It's like a—it's like an avatar or something chasing you that's like—it's called, "Bad influence," influencing you to do drugs or something, and you have to run from it and escape to a portal." (Phase 1, Focus Group 3, Participant 1)

- "...cancer can kind of chase you because—it can catch up to you. It can start to affect different body parts. First, it can probably start off with your brain, your limbs, or your—or your

organs. It can affect any part of your body, because my aunt, she had—she had her heart—she had a heart cancer—she had a heart cancer and she died a couple years ago—rest in peace." (Phase 1, Focus Group 1, Participant 3)

- "A giant cigarette wants you. It's like—and it's running—it's running towards you with a flaming head—a fire. And every time you fight back, it—because you know how they smoke, then they tap it, then the ashes come off. Then when—every time you attack it and the ashes come off, it's getting weaker and weaker. And you know, when you smoke, you get weaker and weaker." (Phase 1, Focus Group 1, Participant 1)

**Pursuit of problems to be solved.** In addition to running away from the storm or avatars, participants recommended that the players chase after problems "on the island to fix things".

- "This is kind of different, I don't know if it would work. But I imagined it as like a map of the island and you would have, like, I don't know, kind of like Carmen Sandiego board game. Where you had to like go to the different spots on the island to fix things. You would have to find an island to model it after, or make one, I guess. But that's sort of how I envisioned it." (Phase 2, Focus Group 2, Participant 4)

- "And then like those same things for the tobacco products and the cancer-causing chemicals. Like they have to go through different steps to figure out what are these things that happened to our island and why are they bad for our island, and then how can we fix it? And then what are the results of getting rid of those products, and how is it helping everyone when they're gone?" (Phase 2, Focus Group 2, Participant 1)

- "I mean 'cause like, well, everything starts out really good, and then it gets bad really quickly, and then you got to try to fix it." (Phase 2, Paired Interview, Participant 2)

**Depiction of tobacco as the enemy.** Some adolescents recommended that the intervention depicts tobacco as the enemy that is chasing the players. This enemy is described as fictional characters (e.g., mutants), natural disasters (e.g., storms), or temptations for tobacco use (e.g., tobacco offers from others).

- "just like tobacco mutants chasing you." (Phase 1, Focus Group 3, Participant 1)

- "maybe it can be like a cigarette storm [chasing you]" (Phase 1, Focus Group 3, Participant 2)

- "You can be running away from bad choices people will try to push you to, like smoking. They'll be like, "Hey, it's the best thing ever. It relieves stress." But then, when you see—because my cousin, he smokes cigarettes. When he doesn't have the cigarette. (Phase 1, Focus Group 1, Participant 1)

**Game obstacles and resources.** Some adolescents recommended game mechanics that support or hinder players' attainment of victory, as they tour the island to save it from tobacco. These may include resources that can protect the player from danger, and time running out was proposed as an obstacle to reaching victory.

- ". . . you have a bunch of things to help you get back to safety and win the game." (Phase 1, Focus Group 3, Participant 1)

- "It's like if you don't complete the challenge—that could be—like—there was the time—we should have the time—so as the storm is closing in, that's the time we're meant to finish the challenge. If we don't finish the challenge, we all just die by the storm. So if you don't—every time you don't—every time you mess up a challenge, the time will speed up, and the storm

will start coming in a little bit faster each and every time you mess up. "(Phase 1, Focus Group 1, Participant 2)

## Discussion

This qualitative research is essential for understanding how to improve tobacco prevention initiatives, particularly in the context of social games for health. Although currently available tobacco prevention programs are successful, there is room for advancement in their design to boost their success and leverage social interaction. In this study, adolescents participated in qualitative research sessions to provide feedback on their preferences and experience with respect to social games. The findings bring to light key components for designing a tobacco prevention program that provides an engaging experience to adolescents. While the uncovered gaming components may not necessarily be unique to tobacco-related games, it is essential to consider adolescents' preferences for these components in the context of a social game that is focused on tobacco prevention. By considering these gaming components, researchers could potentially improve the success of their interventions.

Consistent with the flow state theory [46–48], study participants valued having a balance between challenge difficulty and their ability to overcome a challenge. This balance allows adolescent players to intensely focus and successfully engage, block out distractions, and give complete attention to challenging yet achievable activities. Participants expressed interest in timed activities that require active thinking and hand-eye coordination. For successful engagement, participants shared a need for prior skills and clear game instructions. Ultimately, a gaming balance drives continued achievement that supports effective learning. Future research can explore sophisticated gaming systems that tailor content based on adolescents' skill levels, moving beyond demographic tailoring. Based on the success of tailored digital health programs [49, 50], it is clear that incorporating novel tailoring methods can improve current health education programs.

By supporting both cooperation and competition during gameplay, adolescents emphasized the importance of healthy social interaction. They found the challenge of competition enjoyable and believed it enhances the program. In line with the balance described by the flow state theory, competition adds to the gaming challenges, while cooperation can facilitate adolescents' ability to overcome these challenges. Such social dynamics can play a key role in driving health behavior change. First, in the context of competition, displaying players' progress through leaderboards [51], for example, can boost adolescents' motivation by making their performance visible to others [52, 53]. Similarly, cooperation in a social game can motivate the group to achieve the program objective by leveraging encouragement and shared skills, enhancing the group's self-efficacy for success [54]. Both cooperative and competitive elements of a tobacco prevention program can facilitate a collective motivation to prevent tobacco use [55, 56]. In addition, adolescents' perceived norms about tobacco can shift as they experience a shared objective among peers, to fight tobacco use [53]. While we have not yet applied competition in Storm-Heroes, we anticipate employing a moderate level of competition in the gameplay mechanics.

Aligning with the experiential learning theory that emphasizes exploration of information, adolescents expressed interest in physical engagement and creativity [57, 58]. Previous work has supported the effectiveness of interactive and entertaining tobacco prevention programs [59]. To promote learning and self-efficacy, tobacco education interventions should leverage interactive features of exploration and first-hand discovery. In line with previous research [60], game-based activities can be concise and repetitive, thereby providing prompt feedback

within the interactive setting, to maintaining interest in the health content [60]. Ultimately, this immersive gaming experience can facilitate adolescents' understanding of tobacco risks.

Finally, adolescents expressed a unique perspective concerning the virtual world and its dynamics for tobacco prevention. First, they want to receive ownership of virtual regions as they explore the gaming environment and advance in the game, as it fosters immersion and engagement, in line with the concept of territorial gameplay [61, 62]. Perceived ownership fosters a feeling of belonging and promotes experiential learning by exploring regions and uncovering health information [63]. Participants recommended depicting tobacco as the enemy, allowing adolescents to escape dangerous virtual situations and experiencing the consequences of tobacco use and its threats to the individual and the community. Additionally, adolescents expressed an interest in pursuing and solving problems through gameplay, thereby understanding tobacco risks, lowering their intention to use tobacco, and fostering advocacy for tobacco control and self-efficacy in fighting tobacco within their communities.

## Strengths and limitations

This study identified specific intervention features that best fit the needs of adolescents in the context of a social game for tobacco prevention. Through our qualitative approach, we uncovered key gaming elements that adolescents favor when engaging in a successful learning experience for tobacco prevention. Although the game concept we designed is in its early stages of development, this qualitative work highlighted the design needs within the intervention during Phase 1 and confirmed such design needs when adolescents were presented with the game concept (Storm-Heroes) during Phase 2. During the sessions, teens made it clear that some program elements need to be included. It is also one of the first studies to identify novel design elements for a social game-based program for health promotion, including gaming mechanics (e.g., collaboration and ownership of virtual environments) and activities (e.g., performance-based, and strategy-based activities). These findings identify gaming elements that require deeper conceptualization and assessment in the context of tobacco prevention during subsequent studies.

One limitation of this study was that the female-to-male ratio in each phase was uneven. Nevertheless, our results present qualitative reports from both genders equally, and both genders were randomly distributed among groups, allowing for equal gender representation within groups. Also, the two phases had distinctly different demographics. While Phase 1 was conducted with a sample of adolescents who identified as Black or African-American, Phase 2 included a racially and ethnically diverse sample. It must be noted that Phase 1 involved participant enrollment from a single after-school organization in Texas, while Phase 2 included adolescents scattered throughout Central and Northern Florida. Despite the demographic differences between participants from the two phases, the results show that the two samples were in agreement concerning most of the identified themes. Compared to random sampling, a purposive sample of similar demographic characteristics to the larger adolescent population may have been a better option for Phase 1. The selection process during Phase 1 had to abide by the request of the after-school programs in Texas, whereby only one after-school site was able to participate. While a larger number of participants per phase may have allowed for a more representative sample, a sample of 15 participants in a focus group study is common for qualitative research, and we acknowledge that the reports may not be representative of the entire adolescent population. Having 15 participants in focus group studies can achieve thematic saturation with adolescents, according to earlier studies on qualitative sampling [64, 65]. It must be noted that adolescents' tobacco knowledge may have affected their qualitative reports about tobacco and the game. In this study, we did not quantitatively measure

knowledge about tobacco products. However, knowledge was discussed during the Phase 2 focus groups. Finally, during the Phase 2 sessions, remote discussions did not allow for the study of non-verbal cues and for the activities to be as interactive as they could have been. Nevertheless, remote sessions provided the opportunity for several hard-to-reach adolescents to participate and feel at ease within the comfort of their homes.

Our study is the starting point for further developing Storm-Heroes, an innovative social game for adolescent tobacco prevention. the reported gaming elements are novel to tobacco prevention and education programs. The study also revealed gaming features that could enhance existing tobacco prevention programs. Following these findings, we plan to: (1) further examine the gaming features that we identified; (2) enhance the user experience of our current game concept; and (3) design a final version of the social game-based intervention. For future research, we will use a participatory approach to allow adolescents to take part in the design process, improve Storm-Heroes, and develop health promotional messages that can be incorporated into the program.

## Conclusions

The study results indicate that a game-based social program for tobacco prevention must balance (1) the challenge of gameplay and adolescents' ability to overcome the challenge, (2) collaboration and competition, (3) mental and physical activities, and (4) fighting and escaping challenges. These balances can drive a healthy engagement in the program, thereby facilitating learning. Ultimately, a board game for tobacco prevention is expected to bring adolescents together to create lasting memories that nudge them away from tobacco use and the harm it can cause.

## Supporting information

**S1 Table. COREQ table.** This file presents the table of consolidated criteria for reporting qualitative research (COREQ).
(DOCX)

**S1 File. Instruments.** This file presents the qualitative semi-structured instruments for the study.
(DOCX)

**S2 File. Game description.** This file describes the game in detail.
(DOCX)

## Acknowledgments

Alexander Prokhorov from the University of Texas MD Anderson Cancer Center.

## Author Contributions

**Conceptualization:** Georges Elias Khalil, Jeanie Kim.

**Data curation:** Georges Elias Khalil, Jeanie Kim, David McLean, Erica Ramirez.

**Formal analysis:** Georges Elias Khalil, Jeanie Kim, David McLean.

**Funding acquisition:** Georges Elias Khalil.

**Investigation:** Georges Elias Khalil.

**Methodology:** Georges Elias Khalil, Erica Ramirez, Bairu Zhao, Ramzi G. Salloum.

**Project administration:** David McLean.

**Supervision:** Erica Ramirez.

**Validation:** Bairu Zhao.

**Visualization:** Bairu Zhao, Ramzi G. Salloum.

**Writing – original draft:** Georges Elias Khalil, Jeanie Kim, David McLean, Erica Ramirez, Bairu Zhao, Ramzi G. Salloum.

**Writing – review & editing:** Georges Elias Khalil, Jeanie Kim, David McLean, Erica Ramirez, Bairu Zhao, Ramzi G. Salloum.

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
