## [Decision Letter · Decision Letter 0]

19 Apr 2023

PONE-D-23-01907Identifying adolescents' gaming preferences for a tobacco prevention social game: A qualitative studyPLOS ONE

Dear Dr. Khalil,

Thank you for your patience and submitting your manuscript to PLOS ONE. After careful consideration, we feel that it has merit but does not fully meet PLOS ONE’s publication criteria as it currently stands. Therefore, we invite you to submit a revised version of the manuscript that addresses the points raised during the review process.

Please address all reviewers' comments and change the manuscript accordingly. In addition to the reviewers' comments, I would kindly ask you to follow the PLoS ONE submission formatting for the abstract as well as any other section where you notice a discrepancy.  I much appreciate that you included the overview of how you have addressed the Standards for reporting qualitative research. You have however not included the interview schedule with the questions participants have been asked (22 and 35 questions). You must include these in your revised version. 

We look forward to receiving your revised manuscript.

Kind regards,

Corinne Jola

Academic Editor

PLOS ONE

Journal Requirements:

Reviewers' comments:

Reviewer's Responses to Questions

**Comments to the Author**

1. Is the manuscript technically sound, and do the data support the conclusions?

Reviewer #1: Yes

Reviewer #2: Yes

2. Has the statistical analysis been performed appropriately and rigorously? 

Reviewer #1: N/A

Reviewer #2: Yes

3. Have the authors made all data underlying the findings in their manuscript fully available?

Reviewer #1: Yes

Reviewer #2: Yes

4. Is the manuscript presented in an intelligible fashion and written in standard English?

Reviewer #1: Yes

Reviewer #2: Yes

5. Review Comments to the Author

Reviewer #1: This is a two-phase qualitative study examining adolescents' gaming preferences. In the first one, it aimed to discover the elements of board games that adolescents consider important for a positive learning experience, while the second dealt with confirming these game elements with adolescents, since they are shown a game concept for smoking prevention.

In the introduction, authors should more elaborate the theoretical framework noting that it is very important to include the people for whom the intervention is intended. One of these approaches is that of human-centered design, so I strongly suggest the following literature:

• Chen E, Leos C, Kowitt SD, Moracco KE. Enhancing Community-Based Participatory Research Through Human-Centered Design Strategies. Health Promotion Practice. 2020;21(1):37-48. doi:10.1177/1524839919850557

• UNICEF. Designing Digital Interventions for Lasting Impact. A Human-Centred Guide to Digital Health Deployments. at https://www.unicef.org/innovation/sites/unicef.org.innovation/files/2018-11/unicef_digitalhealthinterventions_final2018.pdf (2018).

From lines 162 to 169, the way in which the number of participants is explained is confusing. Authors should specify in which phase the two participants left the study. Figure 1 needs to be explained more clearly.

Authors should avoid as possible the use of trademarks such as Play-doh, instead they should specify that is a modeling compound to make arts and crafts projects.

The results are confusing to understand. It is my understanding that they were classified by thematic category, so it would be better and clearer to show them by phase.

Reviewer #2: The manuscript looked at a very important topic: preventing adolescents from using tobacco products. The manuscript looked at how games may inform adolescents of tobacco risks and tried to identify key features in social games that are appealing to and engage with the target population. The study is interesting and provides valuable implications for tobacco prevention among adolescents. Below are my comments and suggestions to the authors in the order they appeared to me as I read through their manuscript.

1. The introduction is well-structured and well-written.

2. Have you measured their knowledge about tobacco products?

3. What is the intercoder reliability for the coding?

4. The discussion of the themes identified in focus groups is detailed, which is good. But also consider being more concise.

5. Discussion: With each theme and subtitled themes under it, it seems a lot for readers. I wonder if there is a way to combine some of the subtitled themes under each main theme.

6. Throughout discussing the different themes coming out of the analysis, it is not clear to me how tobacco-related information is waved in. Most discussion seems irrelevant to tobacco and focuses on the programming element. For instance, Figure 2 clearly showed the last major theme is tobacco-related, but not so much for the first three major themes. If that (i.e., social game programming element) is the goal, then I wonder how it is different from other games that are not about health/tobacco.

6. PLOS authors have the option to publish the peer review history of their article (what does this mean?). If published, this will include your full peer review and any attached files.

Reviewer #1: No

Reviewer #2: No

---

## [Author Response · Author response to Decision Letter 0]

16 Jun 2023

Section of the received letter:

Dear Dr. Khalil,

Thank you for your patience and submitting your manuscript to PLOS ONE. After careful consideration, we feel that it has merit but does not fully meet PLOS ONE’s publication criteria as it currently stands. Therefore, we invite you to submit a revised version of the manuscript that addresses the points raised during the review process.

Please address all reviewers' comments and change the manuscript accordingly. In addition to the reviewers' comments, I would kindly ask you to follow the PLoS ONE submission formatting for the abstract as well as any other section where you notice a discrepancy. 

I much appreciate that you included the overview of how you have addressed the Standards for reporting qualitative research. You have however not included the interview schedule with the questions participants have been asked (22 and 35 questions). You must include these in your revised version.

Response:

On behalf of our team, I would like to thank you for your review of this manuscript. We have addressed all reviewers’ comments in this document and made the necessary changes accordingly. We have followed the PLoS ONE submission formatting for the abstract as well as other sections.

In addition to addressing the Standards for reporting qualitative research in the S1 Table (COREQ), we have included the interview schedule with the interview questions in the S1 File, titled “Instruments”. The S1 File presents the 22 and 35 interview questions within a schedule that highlights interview sections and the time allocated to complete each section. We referred to this file within the manuscript on lines 228-230, 240, and 259.

Journal Requirements:

Response:

We have ensured that our manuscript meets PLOS ONE's style requirements.

Response:

We have improved the ethics section of the manuscript, as suggested.

Response:

Thank you for sharing this information. We do wish to make changes to our Data Availability statement. Our circumstances are not covered by the questions asked through the system. Data files are available upon request by contacting the corresponding author and completing a data use agreement with the University of Florida.

Reviewer #1:

This is a two-phase qualitative study examining adolescents' gaming preferences. In the first one, it aimed to discover the elements of board games that adolescents consider important for a positive learning experience, while the second dealt with confirming these game elements with adolescents, since they are shown a game concept for smoking prevention.

Response:

We would like to thank the reviewer for their thoughtful and constructive feedback to improve this manuscript. We answered based on the reviewers’ comments, made the necessary changes within the manuscript, and updated the references as needed. All line numbers refer to the most recent version of the manuscript without tracked changes.

1. In the introduction, authors should more elaborate the theoretical framework noting that it is very important to include the people for whom the intervention is intended. One of these approaches is that of human-centered design, so I strongly suggest the following literature:

• Chen E, Leos C, Kowitt SD, Moracco KE. Enhancing Community-Based Participatory Research Through Human-Centered Design Strategies. Health Promotion Practice. 2020;21(1):37-48. doi:10.1177/1524839919850557

• UNICEF. Designing Digital Interventions for Lasting Impact. A Human-Centred Guide to Digital Health Deployments. at https://www.unicef.org/innovation/sites/unicef.org.innovation/files/2018-11/unicef_digitalhealthinterventions_final2018.pdf (2018).

Response:

Thank you for noting the importance of elaborating on the theoretical framework. We have now added a section describing the study’s theoretical framework (lines 105-113). In short, the social learning theory emphasizes that learning happens through observed, imitated, and reinforced behavior. Social games promote tobacco risk education by encouraging players to emulate healthy behaviors, such as refusing tobacco. Engaging with peers in tobacco prevention content helps adolescents adopt anti-tobacco norms and imitate healthy behaviors.

Thank you for sharing the references on the human-centered design (HCD) approach. We agree. We now presented the HCD approach (lines 115-126). As a summary, we now state that the human-centered design (HCD) approach actively engages adolescents in participatory action research to create solutions. By prioritizing their needs and challenges, we immerse ourselves in their community. Through participatory design, adolescents contribute to a usable health education program for tobacco risk communication. This approach can also involve them in developing a social game for tobacco prevention, similar to the research on designing tobacco graphic warning messages through HCD by Cartujano-Barrera and colleagues (2021).

Cartujano-Barrera F, Azogini C, McIntosh S, Bansal-Travers M, Ossip DJ, Cupertino AP. Developing graphic messages for vaping prevention among Black and Latino adolescents: Participatory research approach. Journal of Participatory Medicine. 2021 Nov 23;13(3):e29945.

2. From lines 162 to 169, the way in which the number of participants is explained is confusing. Authors should specify in which phase the two participants left the study. Figure 1 needs to be explained more clearly.

Response:

We have improved the section that describes the number of participants so that Figure 1 is explained more clearly. We have also clarified that the two participants who left the study did so during Phase 2 (lines 187-194).

3. Authors should avoid as possible the use of trademarks such as Play-doh, instead they should specify that is a modeling compound to make arts and crafts projects.

Response:

Thank you. As advised, we removed any mention of trademarks such as Play-doh. Instead, we now have a simple description of the activity (line 235). This was also updated in Table 1.

4. The results are confusing to understand. It is my understanding that they were classified by thematic category, so it would be better and clearer to show them by phase.

Response:

Thank you for this valuable feedback regarding the classification of themes. We appreciate the suggestion to present the results by phase for better clarity. However, our intention was to follow established qualitative research methods of thematic categorization and merge the data from both phases as a means to validate the results (Creswell and Plano Clark, 2017). This approach enabled us to identify overlaps and agreements between participants from phase 1 and phase 2, thus reinforcing the validity of our findings.

Also, by combining the data from both phases, we aimed to provide a comprehensive understanding of the themes that emerged across the entire study. This strategy has been supported by previous research, as it allows us to capture both breadth and depth in the findings, providing a more robust analysis (Sutton and Austin, 2015; Todres and Galvin, 2005).

Nevertheless, we understand the importance of clarifying this classification of themes in the article. We have made the necessary revisions to the qualitative data analysis section (lines 291-295) to explicitly explain the rationale behind our approach and highlight the merging of data from both phases. We believe these revisions will enhance the clarity and understanding of the results for readers. 

References:

Creswell, J. W., & Plano Clark, V. L. (2017). Designing and conducting mixed methods research. Sage publications.

Sutton, J., & Austin, Z. (2015). Qualitative research: Data collection, analysis, and management. The Canadian Journal of Hospital Pharmacy, 68(3), 226-231.

Todres L, Galvin K. Pursuing both breadth and depth in qualitative research: illustrated by a study of the experience of intimate caring for a loved one with Alzheimer's disease. International Journal of Qualitative Methods. 2005 Jun;4(2):20-31.

Reviewer #2:

The manuscript looked at a very important topic: preventing adolescents from using tobacco products. The manuscript looked at how games may inform adolescents of tobacco risks and tried to identify key features in social games that are appealing to and engage with the target population. The study is interesting and provides valuable implications for tobacco prevention among adolescents. Below are my comments and suggestions to the authors in the order they appeared to me as I read through their manuscript.

Response:

We would like to thank the reviewer for their thoughtful and constructive feedback to improve this manuscript. We answered based on the reviewers’ comments, made the necessary changes within the manuscript, and updated the references as needed. All line numbers refer to the most recent version of the manuscript without tracked changes.

1. The introduction is well-structured and well-written.

Response:

Thank you.

2. Have you measured their knowledge about tobacco products?

Response:

We did not measure knowledge about tobacco products as part of this study. At the beginning of the focus group discussion, we did ask participants in Phase 2 about their knowledge of tobacco products. However, we did not do so through a quantitative survey. Considering that their knowledge about tobacco can affect their qualitative reports about tobacco and the game, we have now added the lack of a quantitative measure of knowledge as a limitation in the discussion section (lines 766-769). 

3. What is the intercoder reliability for the coding?

Response:

The calculation of intercoder reliability has its benefits and disadvantages, as highlighted by a review from O’Connor and Joffe (2020). We chose to omit calculating intercoder reliability based on the objective and approach of the current study. Considering that we applied an iterative human-centered design approach, the intercoder agreement was reached through a qualitative process: The two coders met on a regular basis to discuss the coding process in real-time until full agreement was reached. This method allowed for the discussion of disagreement so they can be resolved, instead of having a single intercoder reliability score with a limited interpretation for program design. We have now added this information under the analysis section (lines 301-306).

Reference:

O’Connor C, Joffe H. Intercoder reliability in qualitative research: debates and practical guidelines. International journal of qualitative methods. 2020 Jan 20;19:1609406919899220.

4. The discussion of the themes identified in focus groups is detailed, which is good. But also consider being more concise.

Response:

We have now made the discussion of the themes more concise in the discussion section, while ensuring that we maintained the conveyed content (lines 683-730).

5. Discussion: With each theme and subtitled themes under it, it seems a lot for readers. I wonder if there is a way to combine some of the subtitled themes under each main theme.

Response:

We understand the concern regarding the abundance of subtitled themes under each main theme. However, it is important to note that the organization and presentation of themes and subthemes are based on the findings of our qualitative thematic analysis. As such, we believe it is best to maintain the distinctiveness of each subtheme without merging them together. This approach allows for a comprehensive exploration of the nuanced aspects within each main theme, providing readers with a thorough understanding of the research findings.

6. Throughout discussing the different themes coming out of the analysis, it is not clear to me how tobacco-related information is waved in. Most discussion seems irrelevant to tobacco and focuses on the programming element. For instance, Figure 2 clearly showed the last major theme is tobacco-related, but not so much for the first three major themes. If that (i.e., social game programming element) is the goal, then I wonder how it is different from other games that are not about health/tobacco.

Response:

Thank you for bringing this to our attention. The goal of the current article is to describe the preferred gaming elements that drive tobacco prevention and education in the context of a social intervention, as opposed to a personal individual-based intervention. As such, adolescents’ reports were not directly aiming to inform tobacco prevention content. Instead, they aimed to convey the most fit gaming strategies for a social intervention. Nevertheless, our discussion section explains how each element that was uncovered in the study relates to improving tobacco-related education. Even though the discoveries in this case may not necessarily be different from those related to regular games, adolescents insisted that their preference would be most appropriate for a social game pertaining to tobacco prevention. In addition, the reported elements have not yet been applied to a game for tobacco prevention. Our next study will highlight the development of tobacco-related messages that have been waved into the gaming intervention. 

Thank you for bringing up this important point. In our research article, the primary focus is to explore the preferred gaming elements that drive tobacco prevention and education within the context of a social intervention, rather than directly addressing tobacco-related information. The insights gathered from adolescents aimed to inform the most suitable gaming strategies for a social intervention, rather than specifically targeting tobacco prevention content.

However, we acknowledge this concern and understand the importance of connecting these gaming elements to tobacco-related education. In our discussion section, we have provided a more explicit explanation of how each identified element can contribute to improving tobacco-related education and prevention efforts. In addition, we have now noted in the discussion section that while the uncovered elements may not necessarily be unique to tobacco-related games, it is essential to consider adolescents' preferences for a social game specifically focused on tobacco prevention (lines 677-680).

Furthermore, it is important to note that the reported elements are novel and have not yet been implemented in an actual tobacco prevention game. Our ongoing research (research article in progress) will address this by presenting our development of tobacco-related messages that have been incorporated into the gaming intervention. We appreciate the reviewer's feedback. We made sure to include this information in the limitation section (lines 775-776).

---

## [Decision Letter · Decision Letter 1]

17 Jul 2023

Identifying adolescents' gaming preferences for a tobacco prevention social game: A qualitative study

PONE-D-23-01907R1

Dear Dr. Khalil,

We’re pleased to inform you that your manuscript has been judged scientifically suitable for publication by two reviewers. It will be formally accepted for publication once it meets all outstanding technical requirements. I informed the team about the required changed regarding data accessibility, however, you may have to complete further forms to ensure these changes have been accepted. 

Kind regards,

Corinne Jola

Academic Editor

PLOS ONE

Additional Editor Comments (optional):

Reviewers' comments:

Reviewer's Responses to Questions

**Comments to the Author**

1. If the authors have adequately addressed your comments raised in a previous round of review and you feel that this manuscript is now acceptable for publication, you may indicate that here to bypass the “Comments to the Author” section, enter your conflict of interest statement in the “Confidential to Editor” section, and submit your "Accept" recommendation.

Reviewer #1: All comments have been addressed

Reviewer #2: All comments have been addressed

2. Is the manuscript technically sound, and do the data support the conclusions?

Reviewer #1: Yes

Reviewer #2: Yes

3. Has the statistical analysis been performed appropriately and rigorously? 

Reviewer #1: Yes

Reviewer #2: Yes

4. Have the authors made all data underlying the findings in their manuscript fully available?

Reviewer #1: (No Response)

Reviewer #2: Yes

5. Is the manuscript presented in an intelligible fashion and written in standard English?

Reviewer #1: Yes

Reviewer #2: Yes

6. Review Comments to the Author

Reviewer #1: (No Response)

Reviewer #2: (No Response)

7. PLOS authors have the option to publish the peer review history of their article (what does this mean?). If published, this will include your full peer review and any attached files.

Reviewer #1: No

Reviewer #2: No

---

## [Editor Report · Acceptance letter]

21 Jul 2023

PONE-D-23-01907R1 

Identifying adolescents’ gaming preferences for a tobacco prevention social game: A qualitative study 

Dear Dr. Khalil:

I'm pleased to inform you that your manuscript has been deemed suitable for publication in PLOS ONE. Congratulations! Your manuscript is now with our production department. 

Kind regards, 

on behalf of

Dr. Corinne Jola 

Academic Editor

PLOS ONE